# Higher cost of finance exacerbates a climate investment trap in developing economies

Nadia Ameli [1✉], Olivier Dessens[1,2], Matthew Winning [1,2], Jennifer Cronin [2], Hugues Chenet [1,3], Paul Drummond[1], Alvaro Calzadilla [1], Gabrial Anandarajah [2] & Michael Grubb [1]

Finance is vital for the green energy transition, but access to low cost finance is uneven as the cost of capital differs substantially between regions. This study shows how modelled decarbonisation pathways for developing economies are disproportionately impacted by different weighted average cost of capital (WACC) assumptions. For example, representing regionally-specific WACC values indicates 35% lower green electricity production in Africa for a cost-optimal 2 °C pathway than when regional considerations are ignored. Moreover, policy interventions lowering WACC values for low-carbon and high-carbon technologies by 2050 would allow Africa to reach net-zero emissions approximately 10 years earlier than when the cost of capital reduction is not considered. A climate investment trap arises for developing economies when climate-related investments remain chronically insufficient. Current finance frameworks present barriers to these finance flows and radical changes are needed so that capital is more equitably distributed.

[1] Institute for Sustainable Resources, University College London, London, UK. [2] UCL Energy Institute, University College London, London, UK. [3] Chair Energy and Prosperity, Risk Foundation Institut Louis Bachelier, Paris, France. ✉email: n.ameli@ucl.ac.uk

A rapid low-carbon transition is central to achieving the well below 2 °C goals of the Paris Agreement[1]. In addition to current policies and plans, meeting current NDC pledges is estimated to require US$130 billion per year of further investment in low-carbon technologies to 2030—an amount which could double or even triple for 1.5-degree consistency[2]. A step-change in the scale and direction of investment is required especially in developing economies, where accessing the appropriate finance is a particular challenge[3], and where many of the impacts of climate change will be most keenly felt[4].

Whether we meet this climate investment challenge will strongly depend on the availability of finance. The geographical distribution of low-carbon finance, defined as capital flows directed towards low-carbon interventions with direct greenhouse gas mitigation benefits[5], is highly unequal. Developed regions are by far the largest recipients, with developing economies—particularly those in Africa—receiving only a small proportion (with the exception of China and, to a lesser extent, India and a few other emerging economies)[5,6].

The capacity to mobilize funding towards low-carbon investment is strongly linked to local enabling environments[7,8]. Differences in macroeconomic conditions, business confidence, policy uncertainties and regulatory frameworks define investment conditions[7,9,10]. In most developing economies, capital markets are immature, not well developed and lack capital stock, making it difficult to access and secure finance[11–13]. This is particularly detrimental for low-carbon projects given their capital-intensive nature compared to traditional fossil fuel assets[8,14,15].

To understand how finance can be mobilised, especially in developing economies, it is crucial to examine local conditions and how they are reflected in investors' perceived investment risks. This is most clearly expressed by the resulting weighted average cost of capital (WACC), which represents the weighted average of the costs of raising funding for a specific project from different sources[16].

There is scarce empirical evidence on the cost of capital for low-carbon assets. This is mainly due to the confidential nature of financing structures behind renewable projects, with underlying financial detail usually not disclosed and difficult to verify[17,18]. Only lately have some studies tried to elicit the WACC for renewable energy projects at country level[19–26]. Overall, these studies found a notably higher WACC in developing economies than in developed ones, but with substantial variation[9,19,27–29]. The discrepancy in WACCs, beyond reflecting investment risks, is also due to more generic factors, including a noted home bias in finance of both individuals and institutions[5,30]—which appears to be even more prevalent for climate finance.

However, most decarbonisation pathway modelling exercises, including those by the International Energy Agency or the International Renewable Energy Agency, do not properly reflect differential financing conditions in their analyses[9,31]. Instead, a (quasi-) uniform cost of capital in the form of hurdle rates is assumed[9]. When more accurate financing costs are used, modelling suggests that the transition to a low-carbon economy in developing economies is more expensive than is usually assumed, while in developed economies the opposite is true[9,31]. Therefore, it is clear that different financing conditions can substantially affect the attractiveness of low-carbon investment in different countries, influencing the pace and the overall cost of the transition in different geographies[14,15,20,27,32,33].

This paper introduces regionally-differentiated WACCs to the TIAM-UCL global energy systems model to assess how these assumptions for the power sector affect patterns of cost-optimal investment for the energy transition consistent with the 2 °C global targets. To the best of our knowledge, this is the first study showing how decarbonisation pathways differ when region-specific financing costs are represented and explored, providing important insights on low-carbon finance for policy-makers.

Energy system transitions in developing economies require particularly high investment, but given their underdeveloped financial markets and domestic risks, investors apply high-risk premiums to the finance they make available, making the transition more expensive than in countries with lower perceived risk. Investments are thus usually foregone, creating a climate investment trap.

A climate investment trap occurs when climate-related investments remain chronically insufficient, due to a set of self-reinforcing mechanisms with dynamics similar to those of the poverty trap[34,35]. High-risk perceptions produce high premiums, increasing the cost of capital for low-carbon investments, thus delaying the energy system transition and the reduction of carbon emissions. Yet, unchecked climate change would lead to greater impacts in these regions[4], affecting production systems and reducing economic output, generating unemployment and political instability, increasing perceived risk even further (Fig. 1).

These dynamics are particularly relevant considering the recent blooming of the sustainable finance (SF) narrative which, through new policy frameworks and practitioner approaches, is expected to overhaul the financial markets' contribution towards the low-carbon economy. While developing economies require the bulk of low-carbon investment, and developed countries are where the most financial capital is, developing economies currently appear to be underserved by the main current SF efforts and initiatives. Understanding the impact of regional differences in the cost of capital underlines the urgency for policy-makers to overcome the climate investment trap as part of their core SF objectives.

The prevalent financing structures used to finance energy infrastructure assets are corporate and project finance[36]. In corporate finance, the sponsoring company secures capital on its balance sheets—meaning that the debt capacity and borrowing costs are based on the company profile—and the related assets are used as collateral in case of default. In project finance, funds are arranged through a special purpose vehicle (SPV), a legally independent company created for each project, where the project's assets and cash flows are offered as primary security. Recent evidence suggests that most energy infrastructure assets (both fossil fuel and renewables) are financed on corporate balance sheets[36,37], despite an increased share of project finance deployed for renewable projects (in 2019 project finance accounted for 35% of the renewable energy asset finance compared to 16% in 2004[36]); 2015 is the only year when the use of project finance for renewables projects exceeded 50%[36].

In our analysis, we made certain simplifications to introduce region-specific WACC values to the TIAM-UCL model. We implement WACC values based only on corporate financing structures due to its predominance, and the complexity in retrieving project finance data for low-carbon assets at the global level (Fig. 2, see WACC section in the "Method"). Corporate finance values reflect WACC estimates at the industry level, hence including the cost of financing for all companies along the value chain (upstream and downstream), which may slightly differ. When comparing our low-carbon WACC values (corporate finance) to WACC values for wind and solar-based on project finance[19], our values are on average slightly lower, but overall trends are similar. Estimations by financing institutions confirm a difference in the order of 100 basis points as a mark-up for project finance compared to corporate finance[38]. Notable exceptions are African countries and Mexico. For Africa, we use an average WACC of approximately 12% for the whole continent, while Steffen[19] reports a solar WACC of 7.8%, 6.6%, 4.2% in Uganda, South Africa and in Zambia, respectively; however, in sharp contrast, Sweerts et al.[27] find that WACC values vary

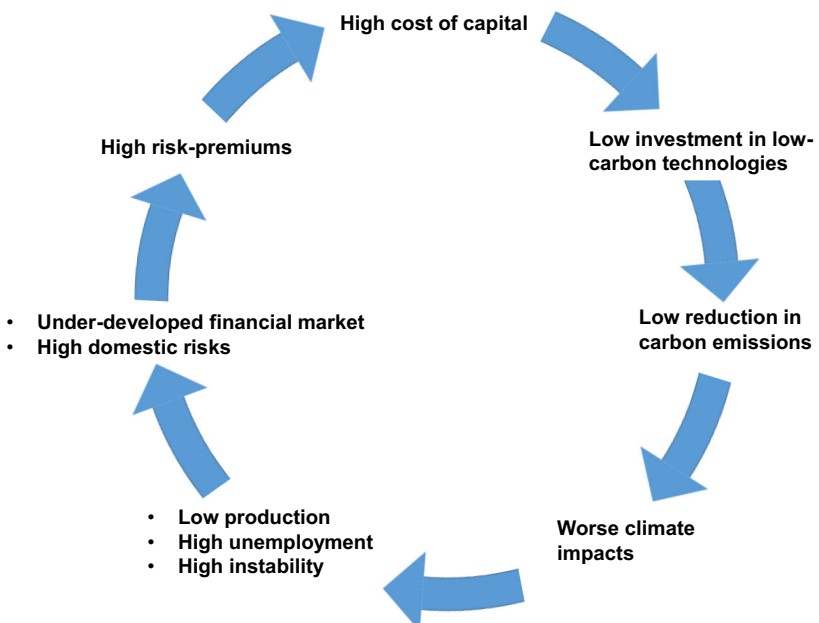

**Fig. 1 The climate investment trap at the macroeconomic level.** The figure shows the set of self-reinforcing mechanisms and related links occurring in developing economies characterised by the high cost of capital. The strength of these links is strongly linked to local conditions implying that the set of self-reinforcing mechanism could be exacerbated (or less relevant) in some economies.

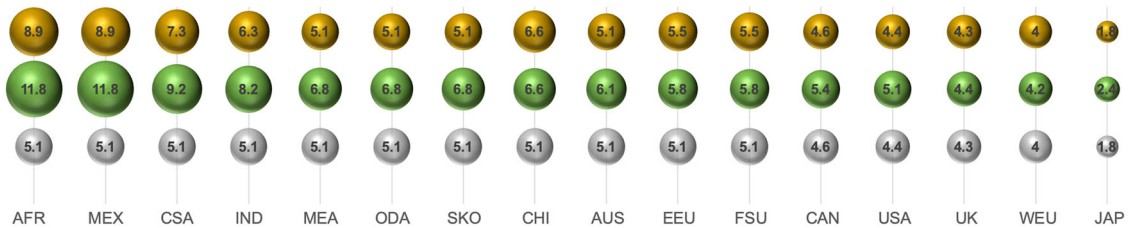

**Fig. 2 Low-carbon, high-carbon and reduced WACC values across the TIAM-UCL regions.** The bubble size and colour reflect the different WACC values for low-carbon (green), high-carbon (brown) technologies, and reduced values (grey). The regions represented are: Africa (AFR), Australia (AUS), Canada (CAN), China (CHI), Central and South America (CSA), Eastern Europe (EEU), Former Soviet Union (FSU), India (IND), Japan (JAP), Mexico (MEX), Middle-east (MEA), Other Developing Asia (ODA), South Korea (SKO), United Kingdom (UK), USA (USA), Western Europe (WEU). For the definition of low-carbon and high-carbon technologies, please refer to Supplementary Table 5.

between 8 and 32% across a sample of 46 African countries. The TIAM model has Africa as one region, and in the face of such diverse estimates, we chose a continent-wide average of 12%. For Mexico, our dataset indicates much higher corporate financing costs (11.8%) compared to project finance (4.9%) for solar projects[19]. In this case, our WACC may underestimate the effect of auctioning systems in reducing financing costs for renewables in Mexico, capturing only the country risk premium[39].

Another assumption relates to the WACC differentials applied, which in our analysis are only captured at country (or regional) level. Recent evidence suggests that WACC varies by other dimensions, particularly technology type[19,40] and investment period[19,41,42]. For instance, a greater risk perception is associated with wind compared to solar assets, due to greater uncertainty surrounding wind resource over solar irradiation[43], and larger operational risks[44]. Time also plays a role in investment risks. By building a successful track record and allowing for learning, technology deployment over time reduces perceived investment risks[20,32,45]. Despite multiple factors explaining WACC differentials, the main source of variation remains in the local context[19], illustrating the importance of capturing geographical variation in the assumed cost of capital.

Given the global scope of the study and the disaggregation to country/regional level, such simplifying assumptions were needed

to derive WACC values at that scale. While differences may exist at project level, the estimated values at the macro level are in line with other estimates in the literature, as discussed above.

The TIAM-UCL (TIMES integrated assessment model) model is used to perform the analysis. TIAM-UCL is a technology-rich bottom-up cost optimisation model that determines the least-cost energy and technology mix that meets future energy demands while respecting technical, economic and policy constraints, such that societal welfare is maximised[46]. As an exogenous input to the model, any changes to the WACC affect annualised technology cost of capital, and subsequently the profile of the cost-optimal energy mix and the total energy system investment requirements. We used a new WACC database covering developed and developing economies[47]. The analysis focuses on the power sector, in the context of the full energy system transition, and accounts for differences in WACC between low-carbon excluding nuclear (hereafter referred to as low-carbon) and high-carbon electricity technologies across regions (Supplementary Table 5).

We examine the financial implications of four scenarios. The first two, GBL and REG, explore the impact of improved representation of regional WACCs in the model. The global scenario (GBL), where WACCs are uniform across regions but differ between low-carbon and high-carbon electricity generation, and are set at the mean global values, weighted by GDP, of 5.9% and

5.1% respectively. The regional scenario (REG) employs the regional WACCs, which vary between low-carbon and high-carbon technologies, as shown in Fig. 2; the WACC is generally higher for low-carbon technologies and in developing economies. In the third and fourth scenarios, FAST and SLOW, the WACC values used in the REG scenario decline by 2050 and 2100, respectively. For each region, the values reduce to whichever is lower—the global average value for high-carbon technologies or the lower WACC of that region (Fig. 2). These scenarios explore the sensitivity to policies that could reduce the WACC over time. Policies, such as credit guarantee schemes, could indeed shift risk away from private investors resulting in lower WACC values[48–50]. Testing specific policies is not part of this exercise, rather we show how investment and electricity generation are affected by WACC reduction over time as a potential outcome of such policies. It is important to note that the FAST scenario achieves identical WACC levels as in GBL in 2050; as such, the results between these two scenarios after 2050 are similar, however, we provide a more realistic pace of change in the FAST scenario as for GBL uses very low WACC values for developing economies from 2020. See the "Method" for more details on the scenarios and the TIAM-UCL model.

We study scenarios that achieve the 2 °C target—rather than the 1.5 °C target—to examine the impact of altering the WACC in scenarios with low reliance on negative emissions technologies, around which there is large uncertainty[51]. We acknowledge that despite this study analyses decarbonisation pathways over the century, WACC values might be surrounded by high uncertainty when considering long-term time horizons.

## Results

We examine how modelling regional WACCs and different speeds of WACC reduction impact electricity decarbonisation pathways and investments in developing and developed economies. To highlight the implications for representative countries with high and low risk profiles, we focus on the results for Western Europe and Africa, which face the principal challenges of replacing high-carbon infrastructure with low-carbon technologies, and of scaling up overall energy supply, respectively. Further, these regions are chosen given their totemic position in the global energy agenda: while Europe aims to be the first climate-neutral bloc in the world by 2050[52], Africa faces rapidly rising energy demand and must leapfrog the use of fossil fuels to meet this demand, and instead deploy clean energy sources if climate targets are to be met[53]. Results for all other regions are reported in Supplementary Information (Supplementary Fig. 1; Supplementary Tables 6 and 7; Supplementary note 1)—emerging economies like Mexico, Central and South America will have similar results to Africa given their WACCs, while India and China will be less affected. Other developed economies will follow very similar paths showed for Europe.

**Representation of global vs regional WACCs.** We start by analysing the implications of implementing regional versus globally uniform WACCs on low-carbon electricity generation and investment requirements (Fig. 3). In 2050 in Africa (panel a), low-carbon electricity generation is much lower when local WACC values are applied (REG) instead of the uniform global value (GBL). This difference remains relatively consistent in absolute term after 2050 (but reduces in relative terms due to the stringency of the 2 °C constraint). However, this produces almost no difference in the investment requirements in low-carbon power (panel b); with undiscounted cumulative low-carbon electricity investment values over the 2020–2070 period of $5.88 trillion and $5.80 trillion for REG and GBL in Africa,

respectively. By 2050, with the same level of investment (approximately $80 billion), and using a uniform WACC, the model projects 35% more low-carbon electricity generation for Africa than when regional values are employed. However, for developed regions the real costs of financing (REG) are similar to the global average (GBL), producing little difference in the resulting pathways (panels c and d).

The impact of regional and globally uniform WACC values on $CO_2$ emissions is greatest for regions with regional WACCs that deviate the most from the uniform WACC. Net-zero emissions in Africa is achieved in 2058 in the GBL scenario and in 2065 in the REG scenario, while the difference is negligible in Western Europe (Fig. 6). Our estimates show that representing the observed local financing conditions leads to regional higher emissions in Africa (+20% in 2050) due to the lower low-carbon investments deployed.

The relevant consequence of implementing regional versus globally-uniform WACC values is that decarbonisation pathways in developing economies are highly affected. Under the REG scenario, they register a much lower (globally cost-optimal) level of low-carbon deployment and a slower rate of emissions reduction than in the GBL scenario.

**Impact of WACC reduction policies.** The potential introduction of policies to reduce low-carbon and high-carbon WACC values has a significant impact on the electricity generation mix in Africa, especially when financing costs are reduced more rapidly (Fig. 4). In 2050, in the FAST and SLOW scenarios, low-carbon electricity production in Africa is 43.1% and 6.5% higher than in the REG scenario, respectively (panel b). Note, the jump seen in the results between 2030 and 2040 in panel b (and in panel b Fig. 5) are inherent to the scenario specifications in our work. Until 2030 the scenarios are aligned to the proposed NDCs in each region. In some regions (mostly developed countries such as WEU) the NDC constraint is stricter than the optimal mitigation needed to achieve the temperature target at the global level. After 2030, only the temperature target applies and the model rescales the regional mitigation levels to the cost-optimal pathway, creating the interregional adjustments seen on the figures for Africa. If cost of capital reductions were brought forward, there would be considerably more low-carbon electricity production in the first half of the century. The REG and SLOW scenarios follow similar pathways until 2050, with the SLOW pathway leading to slightly more low-carbon electricity thereafter, moving toward the levels of low-carbon generation seen in the FAST scenario by 2100. Once again, the relative difference in Western Europe in WACC values and subsequently low-carbon electricity generation is very minor.

The timing of WACC reduction has a large impact on low-carbon power investments in Africa (Fig. 5). Between 2020 and 2070 the cumulative investments in low-carbon electricity are $370 and $310 billion (10 and 9% respectively) more than REG for FAST and SLOW, showing that rapidly lowering the WACC, and the difference between low-carbon and high-carbon technologies, will raise low-carbon power investment in the near term in developing economies (panel a). In 2050, the differential in investment levels between the REG and the FAST scenarios is 15.3% while it starts to diverge only after 2050 in the SLOW scenario. For Western Europe, investments in all scenarios follow a comparable path under similar WACC values. Finally, it is important to mention that these results are partially exacerbated by our initial assumptions on global WACC values (5.1–5.9%). A higher value for the global WACC (for example, applying a standard average WACC of 8%) would most probably reduce the effect on Africa when applying global or regional WACC.

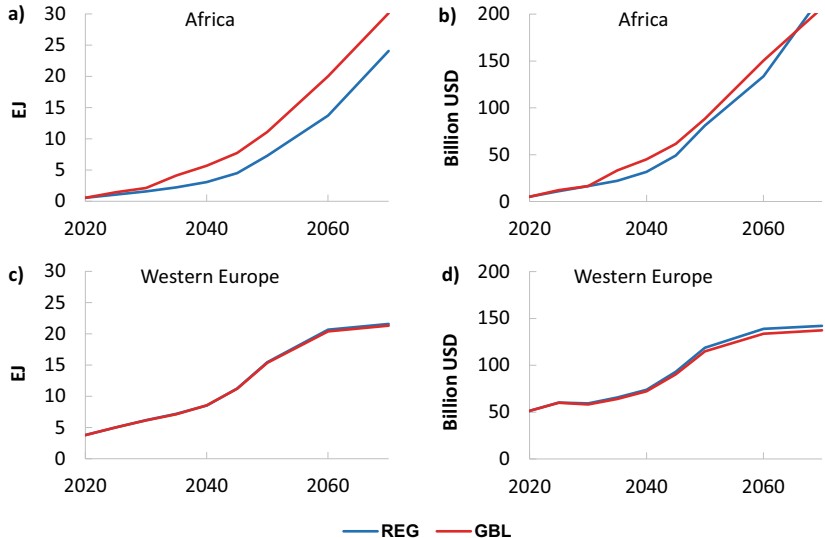

**Fig. 3 Low-carbon electricity generation and related investment per year.** The left-hand panels show low-carbon electricity generation in EJ for (**a**) Africa and (**c**) Western Europe, while the right-hand panels show related investment per year in Billions of USD in (**b**) Africa and (**d**) Western Europe.

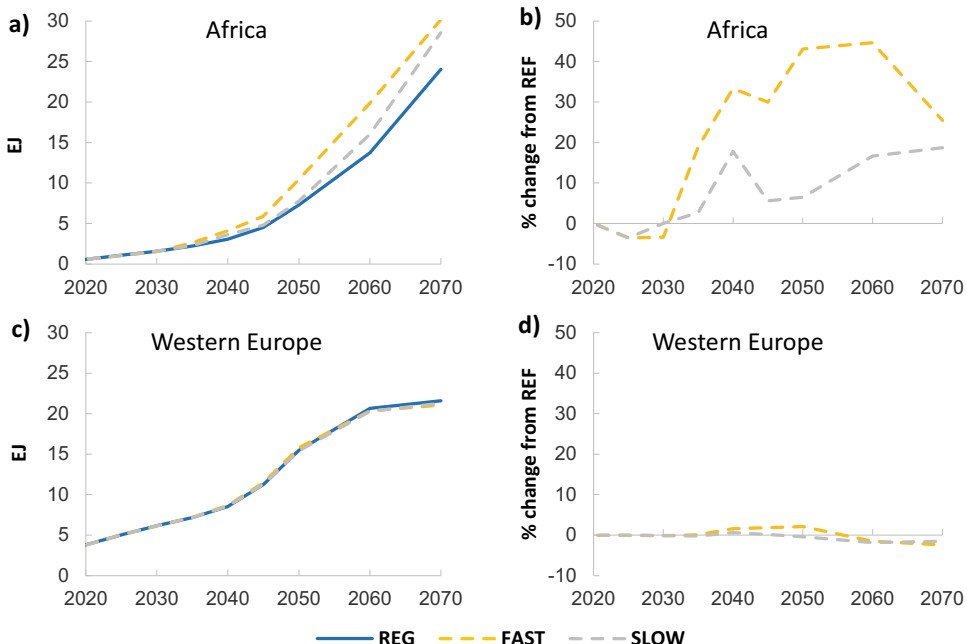

**Fig. 4 Low-carbon electricity generation and percentage changes compared to the REG scenario.** The left hand panels show low-carbon electricity generation in EJ for (**a**) Africa and (**c**) Western Europe, while the right hand panels show the percentage change in generation compared to the REG scenario for (**b**) Africa and (**d**) Western Europe. Investment is constrained to NDCs until 2030 hence impacts of changed financing only emerge after 2030.

Finally, Fig. 6 shows how different WACC reduction rates affect regional $CO_2$ emissions pathways. Again, the most pronounced differences are in regions with the greatest deviations from the global uniform WACC, such as Africa. Net-zero emissions in Africa would be achieved in 2058 for FAST, in 2062 for SLOW and only in 2066 for the REG scenario in a least-cost scenario. As a consequence, slightly higher emissions can be observed in developed countries (Fig. 6 panel b) where mitigation options are getting more expensive, and low WACC values remain unchanged. In all our scenarios, differences in the level and evolution of power sector WACC values do not impact cumulative global $CO_2$ emissions to 2100, as $CO_2$ emissions are constrained by the global temperature limit imposed by model.

Our results suggest that a more rapid WACC reduction will allow developing economies to achieve a much higher level of low-carbon electricity deployment and faster emissions reduction. For Africa, compared to the REG scenario, earlier WACC reduction by 2050 would lead to an almost 50% increase in low-carbon electricity generation by this time, while WACC reduction by 2100 would increase low-carbon electricity generation by 20% in 2100. Lowering the WACC by 2050 (FAST) would also allow Africa to reach net-zero emissions roughly 10 years earlier than in the REG case (Fig. 6 panel a), with higher investments in the near term; a cumulative impact of $430 billion extra low-carbon investment between 2020 and 2050, when compared to the SLOW case.

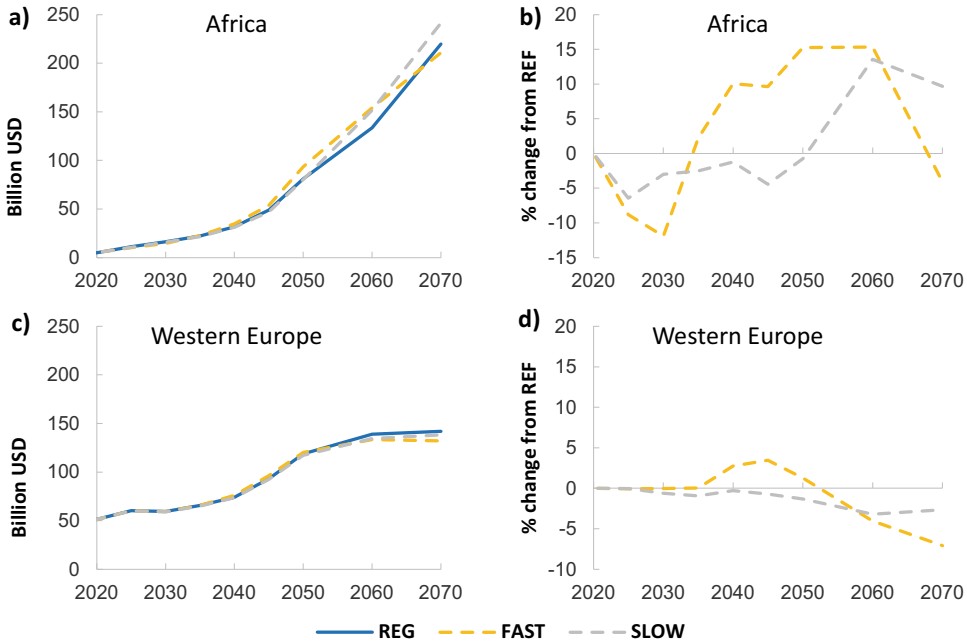

**Fig. 5 Investment levels in low-carbon electricity for the 2 °C scenarios with different WACCs reduction rates.** The left-hand panels show investment levels (Billion USD) in low-carbon electricity for the 2 °C scenarios with different WACCs reduction rates for (**a**) Africa and (**c**) Western Europe. The right hand panels show percentage changes in low-carbon investment compared to the REG scenario for (**a**) Africa and (**d**) Western Europe.

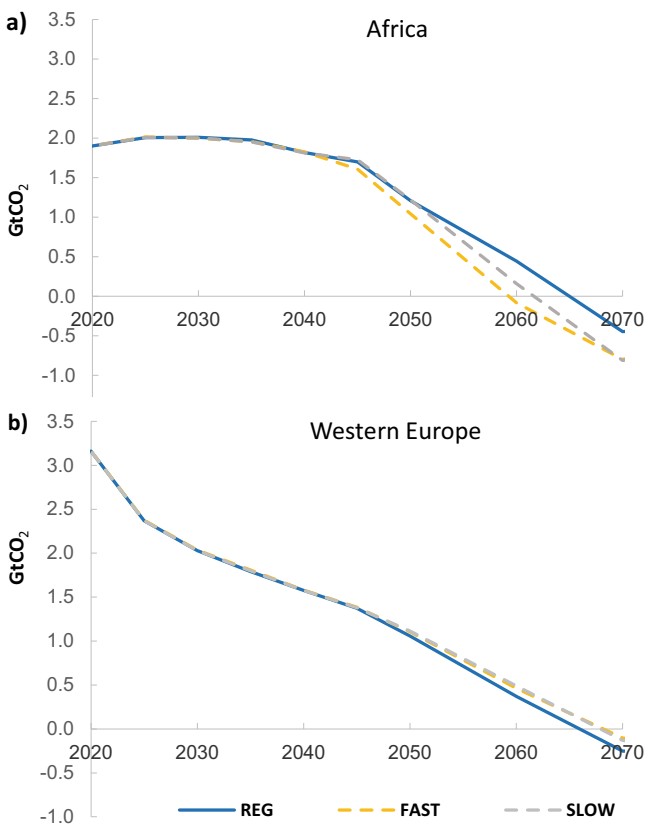

**Fig. 6 CO$_2$ emissions for the 2 °C scenarios with different WACC reduction rates.** The panels show CO$_2$ emissions for the 2 °C scenarios with different WACC reduction rates for (**a**) Africa and (**b**) Western Europe.

## Discussion

The results show that earlier WACC reduction could allow developing regions to reach net-zero earlier. How can earlier WACC reduction be achieved? When considering local financing conditions in developing economies, decarbonisation pathways appear more costly than is often projected. Such economies fall into a climate investment trap when high investment needs are coupled with high WACCs, preventing such investment from taking place. To avoid this, radical changes are needed to facilitate capital allocation towards what investors appear to perceive as high-risk assets and to align current investment pathways with climate targets.

We briefly discuss our results in relation to the existing practices in current SF frameworks[54] both from a local and international perspective and through its main constituents: private capital markets, public and development finance, financial regulation and monetary policy; and how these levers can be used to lower the cost of capital.

The application of environmental social governance (ESG) criteria and risk approaches in investment decision-making can create sustainability links between international private capital markets and developing economies. Nascent SF approaches implementing well below 2 °C targets provide a strong rationale for investors to favour economic activities compatible with net-zero pathways, supporting low-carbon industries over their carbon-intensive counterparts[55–57]. Thus, financial institutions can theoretically influence the greenness of multinational enterprises (MNEs) and infrastructure or industrial projects overseas via equities, bonds, loans[54] or project finance[58,59]. The cost of capital is mainly a result of the perception of future levels of risk and profitability, spanning from reputation issues to real green value and relative financing cost effects, which can be spurred by actions as diverse as ambitious national strategies on low-carbon assets or high-carbon divestment practices. In general, based on existing literature, the sustainability performance of companies tends to lower the cost of capital[60], which would prefigure a virtuous loop with the cost of capital gradually dropping as firms become increasingly present in low-carbon energy.

However, while the type of energy and technology being financed is under acute scrutiny from financial institutions[7], civil society, and governments[61,62], the target country appears to be less of a concern[63]. This tends to indirectly incentivise MNEs' green activities anywhere rather than in specific areas where most needed—for both development needs, and from the strategic standpoint of avoiding the climate investment trap. Moreover, the use of ESG criteria to screen low-carbon investment tends to penalize countries characterised by low democratic, transparency, human rights, and ethical standards, where such criteria are difficult to apply. Similarly, approaches focusing on financial risk, which ranks first among SF practices[55], show high climate-related financial risks in regions that are highly exposed to the physical impacts of climate change—especially if those areas have little capacity to prevent or adapt—and in regions that are carbon-intensive, as a result of expected asset stranding. Consequently, these high-risk regions are left aside by most investors, despite being, again, the most in need of investment[64].

Such limitations do not seem to be addressed by current SF frameworks. For instance, the EU SF action plan[65], which is probably the most ambitious SF policy framework to date, overlooks the impact of financing and investment outside Europe and towards developing economies in general. The Chinese Guidelines for Establishing the Green Financial System[66], the other major SF policy framework, partly addresses this issue by defining how Chinese financial institutions may foster low-carbon finance overseas through green bonds, South-South cooperation and the Belt and Road Initiative. Generally, SF frameworks need to evolve to explicitly target developing economies in how they guide capital flows if they are to play a significant global role.

Given the limited access to international capital markets by developing economies, fostering low-carbon finance requires a strengthening of local financial systems[67], while being careful to avoid the downsides of (over-)financialization[64,68,69]. Supporting the growth of local green bond markets could be a promising way to target low-carbon investment in developing economies, especially if backed by institutional support (and potentially labels) from both local governments and international development banks, which could be involved aside from the private sector in blended finance vehicles[70]. Enhanced green bonds frameworks could also help to overcome some other investment issues linked to the minimum investment size and transaction costs[71]. In addition, public finance and foreign direct investment can help to foster local financial markets by providing initial long-term capital for low-carbon projects[63]. More broadly, public finance, which historically constitutes the main source of climate finance[72] through territorial and technological development, development aid and international development finance, also has a demonstrated capacity in leveraging private capital, and stimulating business models and targeted financial products[63] for low-carbon technologies[42,70,73]. As emphasised by Sharpe and Lenton[74], stronger coordination between donor countries and multilateral development banks could provide a more consistent financial support to target large-scale low-carbon investments instead of multiplying small projects not achieving transformational impact.

Finally, the International Monetary Fund (IMF) certainly can play a core role in facilitating developing countries access to low-carbon finance, whereas macro-financial risks exacerbate sovereign risk and increase the cost of capital[75]. Recent evidence shows that climate vulnerability increases the cost of debt by restricting access to finance[76]. The IMF, by enabling macroeconomic environments through better surveillance of climate-related financial risks and opportunities, enhanced policy supportable to foster synergies between fiscal and monetary policies, and financial assistance that could compensate for the over-exposure to climate risks, could boost resilience and investment in more vulnerable economies[75]. Central banks and financial supervisors are the last big entrants in the SF space[77]. Among them, the first movers were actually in developing economies, where they used their monetary or supervising authority to drive economic development along with climate goals[78–81]. In developed economies, to adhere to their mandates on price stability and systemic risk, such institutions have been more reluctant so far to shift out from their market neutrality principles[55]. Central banks could use their capacity to influence the cost of capital for low-carbon assets, implementing specific policies such as Bangladesh or India's initiatives on green credit allocation and prudential policies[78].

This study shows how developing economies are disproportionately impacted by common assumptions of globally uniform WACC rates. The reality and implications of unequal access to finance across countries are often overlooked in decarbonisation modelling exercises and the pathways they produce. By assuming globally uniform WACC values skewed towards those experienced in developed economies, such exercises underestimate the costs for developing economies of achieving decarbonisation, and thus the climate investment trap they find themselves in. For developing economies, models with uniform power sector WACC rates underestimate the investments required to achieve certain levels of low-carbon electricity generation and overestimate the levels of low-carbon deployment that would be consistent with a globally cost-optimal decarbonisation pathway. Specifically, the TIAM-UCL model

overestimates the low-carbon electricity generation in Africa by 35% when local financing conditions are not considered in 2 °C pathways.

When considering pathways of WACC reduction, our analyses show that low-carbon electricity technologies deploy more quickly when earlier WACC reduction is achieved by 2050 than by 2100. Reducing the cost of capital by 2050 would allow Africa to reach net-zero emissions ~10 years earlier than when reduction is not considered. This in turn implies that earlier action to improve financing conditions could have a significant impact on the speed and timing of the transition.

Radical changes in finance frameworks are thus needed to better allocate capital to the regions that most need it. Elements of SF frameworks currently present barriers to these finance flows. Thus far, SF frameworks have focused mainly on capital in developed markets and do not seem able to significantly address the high cost of capital in developing economies. They should be used to trigger a virtuous circle of reducing the cost of capital, improving access to finance, and increasing rates of investment to avoid a climate investment trap and allow a more even and equitable low-carbon transition to unfold around the world.

The Covid-19 crisis superimposes a huge additional shorter-term obstacle on this challenge. Countries across the world have been severely stressed and required to deploy massive stimulus measures and recovery packages to offset the consequences of the pandemic, which ultimately contribute to higher levels of indebtedness and worsen sovereign risks for many developing economies. These trends, in addition to structural inequalities in financing conditions, further restrict access to finance for the energy transition. Nonetheless, the pandemic may also present a window of opportunity to reframe international market finance, where a lower cost of capital for developing economies would allow for low-carbon development at a more internationally equitable cost. Stronger international policy coordination is critical to enhancing the viability of investments and development globally.

## Methods

**The cost of capital in our analysis.** The WACC is widely used in investment appraisal and decision-making processes[16,82,83]. The WACC allows investors to assess the profitability of different investments representing an appropriate benchmark rate to decide the acceptance or rejection of an investment[84,85].

In this study, the WACC is used as a proxy to assess the differences in risk-premiums associated with energy assets across countries and/or regions, as it represents the weighted average of the costs of raising funding (equity and debt) for a specific investment[16,82]. The cost of equity depends on the risk that equity investors perceive in the project in a specific market, while the cost of debt reflects the default risk that lenders perceive from the same investment in that market[86].

We used a new WACC database covering developed and developing economies as input in the modelling exercise[47]. The data reflects the single variables needed for the WACC calculation according to Eq. (1):

$$WACC = \left(\frac{E}{D+E}\right)*K_e + \left(\frac{D}{D+E}\right)*K_d(1-Tax) \qquad (1)$$

where $E$ is the value of equity, $D$ is the value of debt, $\left(\frac{E}{D+E}\right)$ represents the percentage of equity in the total financing and $\left(\frac{D}{D+E}\right)$ represents the percentage of debt in the total financing, $K_e$ is the cost of equity, while $K_d$ is the cost of debt and finally, $T$ is the tax rate on corporate income.

WACC data at the country level is available for most the European countries (including Austria, Belgium, Bulgaria, Croatia, Czech Republic, Denmark, Finland, France, Greece, Germany, Hungary, Ireland, Italy, Luxembourg, Lithuania, Malta, Netherlands, Portugal, Poland, Slovenia, Slovakia, Spain, Sweden, Switzerland, United Kingdom); the US; China; Australia; Canada; Japan and Mexico. While data at the regional level relates mainly to developing economies, including Latin American countries (including Brazil and Andean, a compound index of Chile, Colombia, Peru), Asian countries (emerging Asian economies as a whole) and global emerging countries (including all other emerging economies except those in Asia and Latin America).

We aggregate WACC country values from Ameli et al.[47] to obtain regional values reflecting the geographical representation in the TIAM-UCL model

(Table 1). We used GDP weighting for regions where we had data for different countries within the same regions (for example WEU or CSA). The same mechanism has been used to calculate the global value used in the GBL scenario. We decided to use GDP weighting as it correlates well with levels of energy investment[87].

**The variables of the WACC.** The cost of capital is computed as the weighted average of the cost of equity and the cost of debt. The risk-free rate and the market risk premium (MRP) adjusted by a beta factor, are the two major building blocks for the calculation of the cost of equity[88]. They reflect the risk premiums requested for investing in a given market/country—where the risk-free rate represents the rate of return that an investor would expect from an asset that is defined as risk-free; and the MRP captures the additional return of a given equity investment when compared to the risk-free rate. The MRP is adjusted by a beta factor to account for the volatility of the asset return in comparison with the market returns as a whole. Finally, the cost of debt reflects the interest rates in the market that an investor would pay, adjusted for the tax-deductibility of interest expenses. While the concept and the WACC equation is commonly accepted, there are a number of different ways to assess its components. Below we describe the single variables needed for the WACC calculation and related data sources (see Ameli et al.[47] for further detail).

*The cost of debt.* The cost of debt is summarised by Eq. (2):

$$K_d(1-Tax) \qquad (2)$$

As a proxy for the cost of debt ($K_d$), we use the average long-term corporate debt yield at a national (when available) or regional area level covering the period July 2015 to July 2016. Data used and sources are reported in Supplementary Table 1.

*Tax rate.* The tax rate (*Tax*) values are based on the KPMG Corporate tax rates dataset updated to July 2016[89].

*Debt-ratio and equity-ratio.* The debt-ratio values and equity-ratio values are based on the Damodaran dataset[90].

*The cost of equity.* We used the capital asset pricing model (CAPM) to assess the cost of equity[91,92]. In the CAPM, the expected return on equity is a linear function of the current risk-free rate and an MRP, scaled by the beta factor (which measures the volatility of the company's assets compared to the whole market's volatility) specific to every company/sector. The cost of equity capital is expressed by Eq. (3):

$$K_e = R_f + \beta(ERP + CRP) \qquad (3)$$

Where $R_f$ represents the risk-free rate, $\beta$ is the beta factor, the ERP is the equity risk premium (ERP) while the CRP is the country risk premium. The ERP plus the CRP determine the overall MRP.

*Risk-free rate.* As a proxy for the risk-free rate, we used the long-term government bond yield of a reference country (considered as a safe country) as presented in Supplementary Table 2. For all European countries we used the 10Y German Bund yield, while for the other regions (including the UK given its closer alignment to the Anglo-Saxon markets than European ones), we employed the 10Y US Treasury Bond yield.

*Beta.* Beta measures the volatility of a security or portfolio compared to the market as a whole. Beta values used in the analysis are based on the Damodaran dataset[90].

*MRP.* The MRP gives an indication of the additional premium to invest in equity assets in a given country and is obtained as a sum of the ERP and the country risk premium.

*ERP.* The ERP used in the analysis is based on the Damodaran dataset[90]. Damodaran follows the historical data approach to forecast long-term equity returns. The ERP is computed using the implied ERP of the S&P 500 calculated against the 10Y US treasury bond from 1928 to 2015. The ERP value used for all countries is 4.45%.

*Country risk premium.* The country risk premium reflects the additional premium associated with investing in the equity market in a specific country. It is computed as a difference between the country's average bond yield and the default-free government's bond yield (a bond whose issuer is highly unlikely to default—for example the German or US treasury bonds)—this difference is defined as the country default spread (CDS). When computing the CDS, some countries (Canada, China and Japan) showed a negative spread and we corrected these negative values by arbitrarily setting the CDS equals to zero. The resulting value is then corrected by an adjusting ratio. The adjusting ratio is the proportion between the standard deviation of the equity market and the standard deviation of the bond market. Table 2 reports the country risk premiums across countries and regions.

**Table 1 WACC aggregation to reflect TIAM-UCL model regional representation.**

| Regions in TIAM-UCL model | WACC dataset[47] |
|---|---|
| Africa | Global emerging countries plus Africa risk premium |
| Australia | Australia |
| Canada | Canada |
| Central and South America | Latin American countries |
| China | China |
| Eastern Europe | Bulgaria, Croatia, Czech Republic, Hungary, Poland, Slovakia, Slovenia |
| Former Soviet Union | Lithuania |
| India | Emerging countries |
| Japan | Japan |
| Mexico | Mexico |
| Middle-East | Asian countries |
| Other Developing Asia | Asian countries |
| South Korea | Asian countries |
| United Kingdom | United Kingdom |
| USA | USA |
| Western Europe | Austria, Belgium, Denmark, Finland, France, Germany, Greece, Ireland, Italy, Luxembourg, Malta, Netherlands, Portugal, Spain, Sweden, Switzerland |

The WACC for Africa is computed as the sum of the WACC in global emerging countries plus an extra country risk premium (5%) to better reflect African countries risk premiums' median values[27].

**Table 2 Country risk premiums across countries and regions.**

| Regions in the TIAM-UCL model | CRP |
|---|---|
| Africa | 8.11% |
| Australia | 1.15% |
| Canada | 0.00% |
| China | 0.00% |
| Central and South America | 4.03% |
| Eastern Europe | 2.80% |
| The former Soviet Union | 1.45% |
| India | 3.11% |
| Mexico | 7.28% |
| Middle-east | 1.76% |
| South Korea | 1.76% |
| Japan | 0.00% |
| United Kingdom | 0.00% |
| USA | 0.00% |
| Other Developing Asia | 1.76% |
| Western Europe | 0.83% |

To derive specific low-carbon and high-carbon WACC values, we relied on the Damodaran dataset[90], which provides beta, equity and country risk premiums, as well as debt-ratio and equity-ratio values at the industry level (namely low-carbon and high carbon sectors); while further detail at technology level is not available.

**Modelling tool**. TIAM-UCL is the TIMES integrated assessment model (TIAM) developed at the UCL Energy Institute[46]. Based on the TIMES model framework[93], TIAM-UCL is a partial equilibrium global multi-region technology-rich bottom-up cost optimisation energy system model. It uses a linear programming approach with the objective to maximise consumer surplus. The model represents energy resource extraction through conversion processes (refineries, electricity and heat generation) and infrastructure to end-users in the residential, commercial, industry, transport and agriculture sectors. Assuming perfect competition with perfect foresight over the modelling period, the model designs a cost-optimal transition of the energy system that meets future service demands, while obeying technical, economic and policy constraints.

In addition to the detailed representation of the energy system, TIAM-UCL was selected for this study due to its bottom-up cost-optimisation paradigm. This approach is better suited to addressing the research questions than other modelling alternatives (such as top-down macroeconomic or simulation models) as investment decisions are made based on costs to determine the cost-optimal technology and resource mix, and as the WACCs can be represented as hurdle rates in order to develop the finance scenarios.

On the resource side, TIAM-UCL represents a total of eleven conventional and unconventional oil resource categories, eight conventional and unconventional gas resource categories, and two coal resource categories. Each category is specified

with an individual supply cost curve for each region. Supplementary Table 3 outlines the key model assumptions on low-carbon technology costs, which are so important for strong mitigation scenarios.

In TIAM-UCL the world is represented as 16 geographic regions: Africa, Australia, Canada, China, Central and South America, Eastern Europe, Former Soviet Union, India, Japan, Mexico, Middle-east, Other Developing Asia, South Korea, United Kingdom, USA, Western Europe (Supplementary Table 4). The regions are linked through trade in crude oil, hard coal, pipeline gas, LNG, petroleum products (such as diesel, gasoline, naphtha, and heavy fuel oil), biomass, and emission permits.

Energy service demands are exogenous inputs to the model; they are projected for the future using drivers such as GDP, population, household size, and sectoral outputs. In this study, the SSP2 shared socioeconomic pathway has been used. The base-year (2005) primary energy consumption, energy conversion, and final consumptions are calibrated to the latest IEA Energy Balance at sector and sub-sector levels. The power generation mix and end-use sector fuel consumption are in line with the historic data (calibrated to 2015 values). In addition to the global social discount rate, various hurdle rates (or WACCs) are used for sector-specific technologies (extraction, transformation, generation or end-use sectors).

To examine the shift between investments in high and low-carbon technologies, we define technologies as shown in Supplementary Table 5. The low-carbon and high-carbon WACCs are differentiated locally between the 16 regions represented in the TIAM-UCL model reflecting the cost of financing in power generation projects.

**Scenarios**. Table 3 summarises the scenarios used in this analysis. We design two sets of scenarios to highlight the financial implications of decarbonisation pathways. The first set includes one scenario implementing regional WACCs (REG) and one scenario using uniform WACCs at mean global values (GBL) differentiating only between the low-carbon and high-carbon generation (5.9% and 5.1% for low-carbon and high-carbon technologies respectively). When using a uniform cost of capital over all regions (GBL), the effect is to reduce the WACC in developing economies compared to their actual cost of financing, while the WACC will increase in developed countries compared to their current values as reflected in the REG scenario. These changes are more evident in the reduction from high WACC values than in the increase from low WACC values to uniform cost, given the WACC deviations in absolute terms. The second set of scenarios considers WACC reduction over different time horizons, namely 2050 and 2100. Considering such pathways of WACC reduction is important to assess how the pace of reducing the cost of capital for low-carbon and high-carbon technologies to uniform values impacts cost-optimal low-carbon electricity generation, power investments and emissions reduction over time. We made simplified assumptions about regions' cost of capital reduction trajectories—we assumed that all countries will experience WACCs reduction at the same pace, without accounting for local structural characteristics that might affect the speed of this process or different trajectories followed by specific groups of countries[94]. The assumption results in faster WACC reduction for developing economies as the WACC gap is more marked in absolute terms in such regions, compared to developed economies (Fig. 2). Figure 7 shows an example of lower WACC for a selected developing economy (Africa) as a function of time (implications for all countries/regions are available in Supplementary Fig. 1, Supplementary Tables 6 and 7, Supplementary note 1).

**Table 3 Scenarios implemented in the TIAM-UCL model.**

| Scenarios | Constraint | WACC |
|---|---|---|
| REG | 2 C in 2100 | Regional (Fig. 2) and constant over the period |
| GBL | 2 C in 2100 | Uniform and constant over the period after 2020 |
| FAST | 2 C in 2100 | Regional differentiation until 2020 (Table 1) linear reduction to 2050 |
| SLOW | 2 C in 2100 | Regional differentiation until 2020 (Table 1) linear reduction to 2100 |

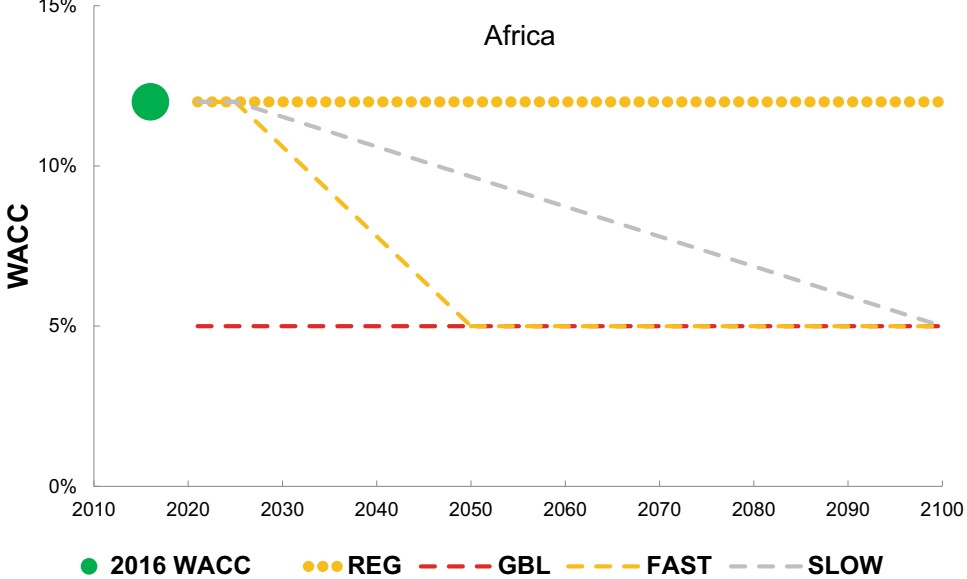

**Fig. 7 Evolution of the low-carbon WACC for Africa under different scenarios.** The figure shows the trajectory of low-carbon WACC for a selected developing economy (Africa) under the four scenarios considered in the analysis.

In all scenarios, GDP growth is calibrated to match the shared socioeconomic pathway SSP2. In addition, energy demand is driven by the SSP2 population growth and the structure of the global economy changes according to the SSP2 projections. All model runs are fixed until 2020 to an unconstrained base run of TIAM-UCL with no climate constraints to represent the rough trajectory of global emissions between 2005 and 2020.

The modelling analysis is based on scenarios achieving a 2 °C target by the end of the century. Overshoot above the temperature limit is allowed in all model runs, meaning that the global temperature rise can exceed 2 °C during the model timeframe but it must return to reach 2 °C or lower in 2100. In all the modelled pathways, the global temperature rise reaches 2.23 °C in 2060.

**Reporting summary**. Further information on research design is available in the Nature Research Reporting Summary linked to this article.

## Data availability
The results data and key source data are provided with this paper. Other datasets used in the determination of the WACC values include the Damodaran dataset (accessed here http://pages.stern.nyu.edu/~adamodar/New_Home_Page/datacurrent.html), the Euro area yield statistics (accessed here https://www.ecb.europa.eu/stats/financial_markets_and_interest_rates/euro_area_yield_curves/html/index.en.html), the US Treasury Yield Curve Rates statistics (accessed here https://www.treasury.gov/resource-center/data-chart-center/interest-rates/Pages/TextView.aspx?data=realyield) and the KPMG Corporate tax rates dataset (accessed here https://home.kpmg/xx/en/home/services/tax/tax-tools-and-resources/tax-rates-online/corporate-tax-rates-table.html).

Other modelling input assumptions are available on reasonable request.

## Code availability
The code underlying the TIAM-UCL model is available at this link: https://doi.org/10.5281/zenodo.3930657.

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

## Acknowledgements
This research was made possible by support from two European Union's Horizon 2020 projects, namely the COP21 RIPPLES (Grant Agreement No 730427) and GREEN-WIN (Grant Agreement No 642018); and the EPSRC as a Standard Research Studentship (Grant number: EP/M507970/1). H.C. acknowledges the support of the *Chair Energy and Prosperity*, under the aegis of the Risk Foundation.

## Author contributions
N.A. coordinated the research. N.A., O.D., M.W., and J.C. wrote the article, with support from P.D., A.C., and H.C. on specific sections. N.A., M.W., O.D., G.A., and A.C. designed the scenarios, with inputs from M.G. O.D., and M.W. led the modelling work, with support from J.C. H.C. contributed to the policy section. All authors reviewed the article.

## Competing interests
The authors declare no competing interests.
