## [Peer Review File · Nature Communications]

REVIEWER COMMENTS

Reviewer #1 (Remarks to the Author):

This study investigates the link between cost of capital (weighted average cost of capital, WACC) and investing in green energy, which is vital to avoid the worst possible scenarios predicted by climate models. This is without a doubt a very important and urgent research topic.

The paper makes an argument about an alleged convergence of WACC among countries; however, it has to be noted that cross-country differences in cost of capital and access to finance tend to be persistent even over longer periods (see comment below). Hence, there is a need for further policy intervention to lower cost of capital, which in turn encourages investment in green energy projects.

The authors refer to international capital flows (presumably foreign direct investment FDI and maybe portfolio investment, which should be clarified) and rightly state that less developed countries only attract a very small fraction of FDI and an even smaller fraction of FDI focused on low carbon projects. It is also correct to argue that macroeconomic and political conditions (e.g., the rule of law) do affect these capital flows.

The study focuses on cost of capital (WACC) to capture the cost of finance. However, access to financial markets is only indirectly measured (see comment below).

The section on the 'climate investment trap' highlights a reinforcing mechanism where high cost of capital reduces investment, which in turn worsens climate outcomes. Climate risk in turn increases cost of capital as shown by Buhr et al. (2018) (as cited) for the country level and recently by Kling et al. (2021) for the firm level.

The study uses the TIAM-UCL model to assess various scenarios of cost of capital and their implications. In summary, this is an interesting study, which provides useful policy recommendations. However, I suggest addressing four main issues as outlined below in a revised version of the paper.

Convergence of WACC

I am not convinced that there is sufficient empirical evidence that countries' cost of capital converge over time. We have witnessed persistent country-specific differences in cost of capital for decades. It would be useful to provide references to empirical studies that justify the selected scenarios.

Financial constraints

Focusing on WACC only works if countries or firms actually have access to finance. Put differently, debt or equity needs to exist to calculate WACC. However, many countries exhibit financial constraints, or they might not have direct access to global financial markets. This is an important limitation of the study. I suggest to consider scenarios where access to finance is restricted.

The role of the IMF

The policy implications derived from this study are certainly relevant. However, I am surprised that the IMF is not mentioned. Arguably, the IMF plays a major role in facilitating access to finance, which has also a significant impact on cost of capital. Hence, IMF interventions reduce financial constraints and also lead to lower cost of capital.

Methods

The section on WACC is well known and covered in standard finance textbooks. This section could be shortened.

However, the modelling tool used for this study is only discussed in a descriptive manner without providing any insights into the mechanics of the simulation. It would be useful to state underlying assumptions and at least provide the main dynamics of the model (e.g. master equation). I checked the 205-page manual on the TIAM-UCL model – but could not find sufficient information to validate the approach. It would be useful if the mathematical model would be made available. It would be even better to have access to the code. There is a very limited literature that explores

the assumptions of the TIAM-UCL model and others in more detail (e.g. Butnar et al., 2020). Hence, it would be important that the authors provide more details on the model and how it was implemented.

Additional references

Butnar et al. (2020) A deep dive into the modelling assumptions for biomass with carbon capture and storage (BECCS): A transparency exercise, *Environ. Res. Lett.* 15 084008

Kling, G., Volz, U., Murinde, V., and Ayas, S. (2021) The impact of climate vulnerability on firms' cost of capital and access to finance, *World Development* 137: 105131

Reviewer #2 (Remarks to the Author):

The paper "A climate investment trap in developing economies" provides a substantial contribution to the literature by implementing more realistic and differentiated WACC in a 'workhorse' integrated assessment model (IAM). By doing so, the authors are able to distil important policy implications for the energy transition. To warrant publication, I believe that the authors should reflect on a couple of substantive issues, which remain unaddressed currently.

The paper introduces regional and technology-specific WACCs. However, it fails to state what types of finance it considers in doing so. There is a vast literature on different types of finance (cf. the most prominent distinction between corporate and project finance), the associated difficulties in estimating WACCs and the caveats that apply. The present manuscript fails to discuss these, indeed veers between one and another (e.g. "bond market" and "funding a specific project" on p. 2). First, the authors should clearly explain how the WACC data is estimated and to what extent these data points are useful for the present analysis. As I understand, the WACC data is estimated from publicly listed companies only, whereas a substantial number of electricity plants worldwide relies on project finance – particularly RE. Second, the authors should explain how the types of finance underlying the estimated WACCs correspond to those needed for the modelled investments and where WACC values can be expected to be off the mark. If indeed the source of the WACC reflects other types of finance than the modelled investments, the authors should use other sources, triangulate values and show scenarios accordingly. In sum, the authors need to demonstrate a better grasp of these differences and a realistic representation in the modelling effort to make a substantial contribution.

The results are driven by the WACC input data. They originate from a WP that is not peer-reviewed. Much more detail is needed on the methods used, the underlying data (type of finance, years considered, technology definitions, regional aggregation or estimation, etc.). Furthermore, some reported regions do not seem to feature in the WP Ameli et al. 2017.

It is possible that I missed an argument but I think the authors should argue why using an IAM is more appropriate for the present analysis compared to other models (i.e. what is the added benefit of having a climate module besides replacing an emissions constraint with a temperature constraint). This relates to the authors' statement on p. 5 that "the analysis focuses on the power sector"; some explanations on this point may be helpful.

TIAM-UCL models energy systems up to 2100. While this may be adequate to quantify climate impacts, I believe it is inappropriate for a WACC analysis. A large strand of research (and recent empirics, cf. financial crisis, COVID-19) has demonstrated that WACC vary substantially over time. I simply think any analysis beyond 2050 misses the point when WACC sensitivities constitute the main purpose. Moreover, it is unclear in many instances why the manuscript discusses 2020-2070 effects (minor point).

In the first two scenarios, the authors compare a global WACC to regional WACCs. First, I believe the authors should add a global WACC scenario as used in the most common energy system models by the IEA or the IRENA. Second, the authors fail to explain why weighing regional WACCs by GDP is a sensible approach to estimate a global WACC. In fact, the key conclusion that Africa is disproportionately affected (cf. first sentence conclusion) seems to be a result of this assumption

because Africa accounts for a disproportionately small share of global GDP. More generally, whatever region deviates most from the assumed global value will show the largest changes, the fact that this is Africa in the present analysis may solely be a consequence of the GDP weights.

In two scenarios, the authors model convergence between technologies. However, in many instances, especially in the reported regions Africa and Western Europe, differences between regions are larger than between technologies. The authors should explain why they expect regional differences to persist through time, whereas technological differences vanish. I would encourage the authors to think about regional convergence scenarios too as I would expect this to have an even larger effect on investment patterns.

Regarding illustrations, I would encourage the authors to show all scenarios in one figure. For example, C2050 seems to be almost identical to GBL for Africa. Something the authors do not discuss and the reader can quickly miss because the lines are in different figures. Furthermore, the manuscript would greatly benefit from a visual illustration in the form of maps. Finally, the authors should explain unexpected patterns in the figures and provide intuitions for those (e.g. 2040 bump in Fig. 4b, pattern in Fig. 5b, etc.). Subpanels are also not identifiable currently (not numbered).

Finally, the discussion focuses on sustainable finance and ESG. Depending on how the authors plan to address the concerns regarding types of finance, the discussion does not fit the paper. Given the prevalence of project finance in the energy sector, I find it unlikely that a discussion on ESG (largely a topic for investment funds, often operating in secondary markets) hits the ball home. Similarly, in the final paragraph on COVID-19, the authors clearly need to make a link to WACCs as there are many implications (cf. public debt levels, central bank actions, etc.).

Smaller points

The paper never mentions that WACCs are more important for the cost of RE compared to FF due to varying capital intensities.

Fig.1: The link to low reduction in carbon emissions seems misleading. The paper uses regionally differentiated WACCs, leading to different low-carbon investment patterns. Whether this leads to a net reduction or increase in emission reductions (and related climate effects) is conceptually unclear. The model has a global warming constraint – hence climate effects are presumably independent of WACC assumptions.

Fig. 2: Year unspecified, technology composition unspecified.

p. 7, 190: Presumably billion instead of million.

Fig. M1: Why are the years before 2020 shown and how should I interpret the drop in GBL in 2020?

Response to reviewers

We thank the reviewers for their insightful comments, which have helped us to improve the manuscript. In particular, we significantly expanded the “cost of capital in our scenarios” section in the main text, as well as the “WACC variables” and “Modelling tool” sections in the method to provide more detail on the cost of capital data and the TIAM-UCL model, along with more elaboration on the policy discussion. Additionally, we amended the manuscript to address the other reviewers’ concerns. Below, we provide our answers point-by-point to reviewers’ comments.

Reviewer 1:

Convergence of WACC

I am not convinced that there is sufficient empirical evidence that countries’ cost of capital convergence over time. We have witnessed persistent country-specific differences in cost of capital for decades. It would be useful to provide references to empirical studies that justify the selected scenarios.

We agree with the reviewer that there is scarce empirical evidence on countries’ cost of capital convergence overtime and this is a strong assumption upon which to motivate scenarios. However, our intention was never to assume that this would occur by itself and instead we aimed to provide a policy analysis to capture the effects of WACC reductions under different policy worlds. We reframe the convergence section as such and we provide an example of a counterfactual “what if” scenario of potential policies lowering capital costs in developing countries. Policies, such as credit guarantee schemes, could indeed shift risk away from private investors resulting in lower WACCs. Testing specific policies is not part of this exercise, rather we show how investment and electricity generation are affected by WACC reduction overtime. Therefore, the FAST and SLOW scenarios are illustrating how policies that attempt to improve access to finance in developing countries for renewable technologies, can act generally to assist climate mitigation.

Additional note

Please note, in this revised version of the manuscript, we have sought to clarify substantially the purpose of the two sets of scenarios that we modelled. In doing so, we have renamed the second set, in which the WACC is lowered over two different timeframes, from C2050 and C2100 to FAST and SLOW. We believe this makes their purpose and comparison clearer, and also prevents any potential confusion between the scenario name and the years themselves.

Financial constraints

Focusing on WACC only works if countries or firms actually have access to finance. Put differently, debt or equity needs to exist to calculate WACC. However, many countries exhibit financial constraints, or they might not have direct access to global financial markets. This is an important limitation of the study. I suggest to consider scenarios where access to finance is restricted.

We thank the reviewer for this comment. In the introduction, we emphasised that developing economies have limited/restricted access to finance and how this is a challenge to support low-carbon investment. From a modelling perspective, including scenarios where access to finance is restricted, would further raise WACC values, which we believe is already captured in our current WACC differentials.

The role of the IMF

The policy implications derived from this study are certainly relevant. However, I am surprised that the IMF is not mentioned. Arguably, the IMF plays a major role in facilitating access to finance, which has also a significant impact on cost of capital. Hence, IMF interventions reduce financial constraints and also lead to lower cost of capital.

We appreciate the reviewer's suggestion on the IMF, which indeed could play a relevant role in facilitating access to finance and lowering financing costs. We have made specific reference to IMF's role in enabling macroeconomic environments in developing economies, whereas macro financial risks worsen sovereign risk and increase the cost of capital. Additionally, we have further elaborated on the role of multilateral development banks and added new references to support the whole discussion.

Methods

The section on WACC is well known and covered in standard finance textbooks. This section could be shortened.

To accommodate the reviewer's request to shorten the WACC section, we streamlined the first part of "The cost of capital" description. Overall, however, we substantially expanded the WACC section by adding a new sub-section on WACC variables to address the second reviewer's concern on the data used, coverage and geographical aggregation.

However, the modelling tool used for this study is only discussed in a descriptive manner without providing any insights into the mechanics of the simulation. It would be useful to state underlying assumptions and at least provide the main dynamics of the model (e.g. master equation). I checked the 205-page manual on the TIAM-UCL model – but could not find sufficient information to validate the approach. It would be useful if the mathematical model would be made available. It would be even better to have access to the code. There is a very limited literature that explores the assumptions of the TIAM-UCL model and others in more detail (e.g. Butnar et al., 2020). Hence, it would be important that the authors provide more details on the model and how it was implemented.

We thank the reviewer for this comment and apologise for the lack of detail on the TIAM-UCL model. To address this concern, we have improved the model description in the Annex, added references to the TIMES platform documentation, and updated documentation for TIAM-UCL model. We have also improved the text to explain why TIAM-UCL was chosen over other models (See response to Reviewer 2).

Reviewer 2:

Types of finance

The paper introduces regional and technology-specific WACCs. However, it fails to state what types of finance it considers in doing so. There is a vast literature on different types of finance (cf. the most prominent distinction between corporate and project finance), the associated difficulties in estimating WACCs and the caveats that apply. The present manuscript fails to discuss these, indeed veers between one and another (e.g. "bond market" and "funding a specific project" on p. 2).

We thank the reviewer for this comment and apologise for the lack of clarity around the types of finance considered in the analysis. We substantially expanded the section on "The cost of capital in our scenarios" to better reflect current knowledge on financing structures used to finance renewable technologies. In the introduction section, we also reinforced the difficulties

linked to the estimation of WACC values and removed any conflicting sentences mixing the types of finance (e.g. bond market in Africa). Finally, we added quite a few new references to make the whole section more robust and better reflect existing findings from the literature.

First, the authors should clearly explain how the WACC data is estimated and to what extent these data points are useful for the present analysis. As I understand, the WACC data is estimated from publicly listed companies only, whereas a substantial number of electricity plants worldwide relies on project finance – particularly RE.

To provide more clarity around WACC estimates, we substantially expanded the WACC section in Supplementary Information. We added a new sub-section on how WACC variables have been computed, data used, coverage and geographical aggregation. Moreover, in the main text (section “The cost of capital in our scenarios”) we clearly state that our study is based only on corporate finance estimates and mention the simplified assumptions made to facilitate the analysis. Recent evidence suggests that most of energy infrastructure assets (both fossil fuel and renewables) are actually financed on balance sheets (WEI 2020, FS-UNEP 2020), despite an increased share of project finance deployed for renewable projects (in 2019 project finance accounted for 35% of the renewable energy asset finance compared to 16% in 2004, FS-UNEP 2020); 2015 is the only year when the use of project finance for renewables projects exceeded 50% (FS-UNEP 2020).

References:

World Energy Investment (2020) - May 2020. IEA, Paris.

FS-UNEP (2020). GLOBAL TRENDS IN RENEWABLE ENERGY INVESTMENT 2018 - UN Environment, the Frankfurt School-UNEP (FS-UNEP).

Second, the authors should explain how the types of finance underlying the estimated WACCs correspond to those needed for the modelled investments and where WACC values can be expected to be off the mark. If indeed the source of the WACC reflects other types of finance than the modelled investments, the authors should use other sources, triangulate values and show scenarios accordingly. In sum, the authors need to demonstrate a better grasp of these differences and a realistic representation in the modelling effort to make a substantial contribution.

We thank the reviewer for this comment. In our analysis, we made a few simplifications to introduce WACC values to the TIAM-UCL model. Such assumptions (now clearly explained in the section “The cost of capital in our scenarios”) relate to the WACC values used, based only on corporate finance, and WACC differentials applied, captured at country (or regional) level.

We believe that these simplifications do not affect the implications of our analysis. We implement WACC values based only on corporate financing structures as this is the predominant way to finance energy infrastructure assets. Regardless, when comparing our renewable WACC values (corporate finance) to WACC values for wind and solar based on project finance (Steffen 2020), our values are on average slightly lower, but overall, trends are similar. Estimations by financing institutions confirm a difference in the order of 100 basis points as a mark-up for project finance compared to corporate finance (Roland Berger 2011). Notable exceptions are African countries and Mexico. For Africa, we use an average WACC of approximately 12% for the whole continent, while Steffen (2020) report a solar WACC of 7.8%, 6.6%, 4.2% in Uganda, South Africa and in Zambia, respectively. However, we feel a higher WACC for the African continent as a whole to be appropriate – Sweerts et al. (2019) suggest that WACC values vary between 8% and 32% across a sample of 46 African countries. For Mexico, our dataset suggests much higher corporate financing costs for solar projects (11.8%) compared to project finance (4.9%) (Steffen 2020). In this case, our WACC

may underestimate the effect of auctioning systems in reducing financing costs for renewables in Mexico, capturing only the country risk premium.

Another simplified assumption relates to the WACC differentials applied, which in our analysis is only captured at country (or regional) level, while recent evidence suggests that WACC varies also among other dimensions, namely technology type and investment period. However, despite the occurrence of multiple factors explaining WACC differentials, the main source of capital cost variation remains in the local context, illustrating the importance to capture country investment conditions in capital costs.

Given the global scope of the study and the disaggregation to country/regional level, such assumptions were needed to derive WACC values at that scale. While differences may exist at project level, the estimated values at the macro level are in line with other estimates in literature, as discussed above.

References:

Roland Berger (2011). The structuring and financing of energy infrastructure projects, financing gaps and recommendations regarding the new TEN-E financial instrument. Final report, Tender No. ENER/B1/441-2010.

Steffen B. (2020). Estimating the Cost of Capital for Renewable Energy Projects. *Energy Economics*, 88, 104783.

Sweerts, B., Dalla Longa, F., van der Zwaan, B. (2019). Financial de-risking to unlock Africa's renewable energy potential. *Renewable and Sustainable Energy Reviews*, 102, 75-82.

WACC input data

The results are driven by the WACC input data. They originate from a WP that is not peer-reviewed. Much more detail is needed on the methods used, the underlying data (type of finance, years considered, technology definitions, regional aggregation or estimation, etc.). Furthermore, some reported regions do not seem to feature in the WP Ameli et al. 2017.

We thank the reviewer for this comment and apologise for the lack of detail behind our WACC calculations. To respond to the reviewer request, we substantially extended “The cost of capital” part in the method section. We added a new section entitled “The variables of the WACC” which reports the single variables behind the WACC calculation and related data sources. Detail is also provided on the data timeframe and regional aggregation used to input the WACC data into TIAM-UCL model; while a table summarising technology classification has been added in the “modelling tool” section.

Finally, it is important to mention that despite the WACC data input originates from a working paper that is not peer-reviewed, that analysis has been reviewed internally by two project members not involved in the study, neither belonging to the UCL team.

Appropriateness of IAM models

It is possible that I missed an argument but I think the authors should argue why using an IAM is more appropriate for the present analysis compared to other models (i.e. what is the added benefit of having a climate module besides replacing an emissions constraint with a temperature constraint). This relates to the authors' statement on p. 5 that “the analysis focuses on the power sector”; some explanations on this point may be helpful.

We thank the reviewer for the suggestion and have added text to explain the choice of model in the Method section. In addition to the detailed representation of the energy system, TIAM-UCL was selected for this study due to its bottom-up cost-optimisation paradigm. This approach is better suited to addressing the research questions than alternatives (such as top-down macroeconomic or simulation models) as investment decisions are made based on costs to determine the cost-optimal technology and resource mix, and because the WACCs can be represented as hurdle rates in order to develop the finance scenarios. Further, a strength of using TIAM-UCL is that it allows analysis of the power sector in the context of the full energy system transformation; we added a comment to this effect on page 7.

Analysis timeframe

TIAM-UCL models energy systems up to 2100. While this may be adequate to quantify climate impacts, I believe it is inappropriate for a WACC analysis. A large strand of research (and recent empirics, cf. financial crisis, COVID-19) has demonstrated that WACC vary substantially over time. I simply think any analysis beyond 2050 misses the point when WACC sensitivities constitute the main purpose.

We understand the reviewer's concern about the analysis timeframe and WACC variation overtime, as the time dimension plays a role in investment risks. We have now clearly mentioned this aspect as an assumption in our analysis (section "The cost of capital in our scenarios"). However, as the aim of this paper is to examine the impact of representing regionally specific WACC values within long-term decarbonisation pathways, we prefer to frame the analysis within our current timeframe. Scenarios consistent with 2°C warming reach net zero around 2070, with many developing countries undertaking significant mitigation between 2050 and 2070 (ref IPCC AR5), so it is appropriate to study the role of finance over this timeframe, as well as focussing on the near term. We therefore use an energy system model which is parameterised for these longer-term pathways and whose climate and carbon constraints have to be defined over a time horizon that stretches towards the end of the century.

Moreover, it is unclear in many instances why the manuscript discusses 2020-2070 effects (minor point).

We thank the reviewer for this comment. The period discussed has been chosen to overlook the next 50 years (2020-2070) when global net-zero emission is achieved under the 2C temperature target. We have clarified this aspect in the main text ("the cost of capital in our scenarios" section).

Scenarios

In the first two scenarios, the authors compare a global WACC to regional WACCs. First, I believe the authors should add a global WACC scenario as used in the most common energy system models by the IEA or the IRENA. Second, the authors fail to explain why weighing regional WACCs by GDP is a sensible approach to estimate a global WACC. In fact, the key conclusion that Africa is disproportionately affected (cf. first sentence conclusion) seems to be a result of this assumption because Africa accounts for a disproportionately small share of global GDP. More generally, whatever region deviates most from the assumed global value will show the largest changes, the fact that this is Africa in the present analysis may solely be a consequence of the GDP weights.

We thank the reviewer for this comment. Relating to the suggested addition of a "global WACC scenario as used in the most common energy system models" we believe our current GBL scenario already covers this for the purpose of our analysis and do not believe a further

scenario would add extra value. In our “GBL” scenario all countries have the same WACCs; specifically, the WACCs are 5.9% and 5.1% for green and brown power technologies in all TIAM regions. Adding another Global scenario of the kind requested would require further aggregation of the data and another step in the process, in order to be consistent with the underlying Ameli et al (2017) WACC values i.e. the country level data would need to be weighted to estimate a global WACC and then weighted again to balance green and brown. Also, we believe this would add additional complication for readers in navigating scenarios, especially when the focus of the paper is on the differences between (i) country assumptions (See the results described in ‘Global vs Regional WACCs’) and (ii) policy assumptions (See ‘Impact of WACC reduction policies’) but not about the difference between renewables vs fossils. Therefore, overall, our feeling is that an additional scenario would detract from the focus of the paper and overcomplicate matters.

The choice of weighting WACC values by GDP was necessary to reconcile country/regional WACC values computed in Ameli et al 2017 with the regional representation in the TIAM-UCL model. The weighting was needed mostly for the European and South America regions where we had data for different countries within these areas (e.g. Western Europe in the TIAM-UCL model includes countries with very different WACC values, such as Germany and Greece). To produce a composite WACC value for Europe we decided to weight the national WACC by their respective GDP. The same mechanism has been used to calculate the global value used in GBL scenario. We decided to use GDP weighting for calculating composite WACC as it is an appropriate choice for an indicator of economic activity. Moreover, GDP correlates with levels of energy investment more closely than population, for example.

We substantially extended “The cost of capital” part in the method section to report the regional aggregation to input WACCs data into TIAM-UCL model.

Convergence

In two scenarios, the authors model convergence between technologies. However, in many instances, especially in the reported regions Africa and Western Europe, differences between regions are larger than between technologies. The authors should explain why they expect regional differences to persist through time, whereas technological differences vanish. I would encourage the authors to think about regional convergence scenarios too as I would expect this to have an even larger effect on investment patterns.

We thank the reviewer for this comment. Convergence hypotheses are strong assumptions upon which to motivate scenarios, especially as noticed by the reviewer, regions or countries may experience larger economic differences than technological ones. In presenting the convergence scenarios, our intention is to provide a policy analysis to capture the effects of WACC reductions under different policy worlds. We have clarified the framing of the convergence section as such and we provide an example of a counterfactual “what if” scenario of potential policies lowering capital costs in developing countries. Policies, such as credit guarantees schemes, could indeed shift risk away from private investors resulting in lower WACCs. Testing specific policies is not part of this exercise, rather we show how the least-cost scenarios of electricity generation and related investments are affected by WACC reduction overtime. Therefore, the FAST and SLOW scenarios illustrate how policies that attempt to improve access to finance in developing countries for renewable technologies can act to assist climate mitigation.

Additional note

Please note, in this revised version of the manuscript, we have sought to clarify substantially the purpose of the two sets of scenarios that we modelled. In doing so, we have renamed the second set, in which the WACC is lowered over two different timeframes, from C2050 and

C2100 to FAST and SLOW. We believe this makes their purpose and comparison clearer, and also prevents any potential confusion between the scenario name and the years themselves.

Charts

Regarding illustrations, I would encourage the authors to show all scenarios in one figure. For example, C2050 seems to be almost identical to GBL for Africa. Something the authors do not discuss and the reader can quickly miss because the lines are in different figures. Furthermore, the manuscript would greatly benefit from a visual illustration in the form of maps.

We thank the reviewer for this suggestion to combine all the scenarios in one figure. We have considered it carefully but decided to keep the figures as they are. This is because the first set of scenarios examines an improvement to the model, while the second set explores the sensitivity to potential policy interventions. The purposes of the two sets of scenarios has been clarified in the main text.

We don't believe a map showing results at regional level would add value to the analysis given the geographical representation in the TIAM-UCL model, where some regions are represented as individual countries, while other regions aggregate many countries (e.g. South Korea vs South America). However, a map of the regions represented in the model is provided in the TIAM-UCL documentation, an updated version of which is now referenced in the paper.

Finally, the authors should explain unexpected patterns in the figures and provide intuitions for those (e.g. 2040 bump in Fig. 4b, pattern in Fig. 5b, etc.).

The "unexpected" patterns are mostly seen in the ratio of change when comparing scenario results to the reference case (REG), and they occur around the year 2030. These patterns are inherent to the scenario specifications in our work. Before 2030 the scenarios are constrained by the regional level of proposed NDC; each region has a "GHG emission maximum" that needs to be achieved. In some cases the NDC is stricter than the optimal mitigation needed to achieve the temperature target.. This is the case for Western Europe. After 2030, only the temperature target applies and the model can rescale the regional mitigation level to the cost optimal pathway (i.e. the model reduces the effort in Western Europe and increases the mitigation in Africa in our case), creating the interregional readjustments seen on the figures. We clarify these aspects, we expanded the main text accordingly ("Results" section).

Subpanels are also not identifiable currently (not numbered).

We thank the reviewer for this suggestion. The figures have been modified and the subpanels' identification used in the text when required.

Policy section

Finally, the discussion focuses on sustainable finance and ESG. Depending on how the authors plan to address the concerns regarding types of finance, the discussion does not fit the paper. Given the prevalence of project finance in the energy sector, I find it unlikely that a discussion on ESG (largely a topic for investment funds, often operating in secondary markets) hits the ball home.

We agree with the reviewer that ESG approaches and sustainable finance were initially more focused on asset management and then on banking. Nevertheless, those related criteria, tools and practices now touch upon the whole financial system perimeter as showed and reported by recent evidence. We added several explanations and references to make this point more explicit and clearer.

Similarly, in the final paragraph on COVID-19, the authors clearly need to make a link to WACCs as there are many implications (cf. public debt levels, central bank actions, etc.).

We thank the reviewer for this suggestion. We have revised the Covid-19 paragraph stressing the link between stimulus packages/recovery plans, higher levels of indebtedness and worsen sovereign risks among many developing economies. These trends, in addition to structural inequalities in financing conditions, further restrict access to finance for the energy transition. We conclude the paragraph by calling for a more coordinated policy action to enhance the viability of investments and development globally.

Smaller points

The paper never mentions that WACCs are more important for the cost of RE compared to FF due to varying capital intensities.

We thank the reviewer for this comment. We have clarified in the introduction that renewable technologies are inherently more sensitive to changes in the WACC than traditional fossil fuel assets given their capital-intensive nature coupled with low operational costs.

Fig.1: The link to low reduction in carbon emissions seems misleading. The paper uses regionally differentiated WACCs, leading to different low-carbon investment patterns. Whether this leads to a net reduction or increase in emission reductions (and related climate effects) is conceptually unclear. The model has a global warming constraint – hence climate effects are presumably independent of WACC assumptions.

We thank the reviewer for this comment. The global emission levels and pathways are indeed determined in our scenarios by the temperature target (equivalent to a carbon budget) and the global cost of mitigation (itself depending on the regional technological costs and WACCs). However, WACC values have an important impact on the distribution of that global constraint at a regional emissions level. e.g. lower WACC in Africa in the policy scenario drives lower regional emissions (fig 6). As a consequence, slightly higher levels of emissions can be observed in developed countries where mitigation options are getting more expensive and pre-existing low WACC values are not changing under policy scenarios (as explained in the scenario description: “Different WACC reduction rates”. We clarified these aspects in the main text (“Results” section).

Fig. 2: Year unspecified, technology composition unspecified.

We thank the reviewer for this comment and apologise for the lack of detail behind our WACC calculations. To respond to the reviewer request, we substantially extended “The cost of capital” section in the methods. Detail is provided on the data timeframe, while a table summarising technology classification has been added in the “modelling tool” section.

p. 7, 190: Presumably billion instead of million.

We thank the reviewer for spotting this mistake; it has been corrected in the text.

Fig. M1: Why are the years before 2020 shown and how should I interpret the drop in GBL in 2020?

We thank the reviewer for this comment. All scenarios are optimised under the mitigation target and mitigation is starting only after the year 2020. The energy system in 2020 is fixed to represent present day situation. The chart also captures the WACC values extracted from Ameli et al 2017 which represents cost of capital in 2016. We have modified the figure to show the difference between 2016 WACC values from Ameli et al 2017 and the post-2020 behaviour in the different scenarios in the TIAM-UCL model.

REVIEWERS' COMMENTS

Reviewer #1 (Remarks to the Author):

The authors addressed all my concerns. The revised paper provides a lot more detail on the methodology, which is appreciated. I do not have any additional comments. In summary, the revised paper should be accepted for publication.

Reviewer #2 (Remarks to the Author):

The authors have provided a substantially revised manuscript. I particularly appreciate the new methods section, which gives very helpful detail about the WACC calculation, the model choice and the implementation in TIAM-UCL.

In response to the authors' answer to "WACC input data", I commend the authors for the added clarity regarding WACC data and the type of finance. The methods now clearly lay out how WACC values are derived. However, unless I missed it, there is still a crucial point missing. I would encourage the authors to explain how the green and "brown" WACC values were derived. Table M7 makes the technology classification in TIAM-UCL transparent, which is great. A similar table would be helpful for the derivation of the WACCs, i.e. what sectors were used to estimate betas etc. to obtain the green and "brown" values.

In response to the authors' answer to "Charts". I understand the argument against showing all scenarios in one figure. However, I think the observation that GBL and FAST yield (by definition) very similar results for Africa should be stated in the manuscript. This is an interesting point to discuss as it means that models using uniform rates may unintentionally assume ambitious policies. Furthermore, I would still be interested in an explanation (or an intuition) for the bump in Figure 4b, which is in 2040. The authors have provided an intuition for a 2030 effect, but I think this does not explain what we see in the figure.

I can understand the authors' argument for weighing values by GDP when aggregating national or regional WACCs. But by definition, this approach leads to a large difference for Africa when comparing GBL to REG and a small one for Western Europe because the former WACC input values "count less" than the latter in calculating the global average. Hence, to some extent, the conclusions the authors reach are driven by this weighing assumption, which is something that needs to be stated clearly. If, for example, the authors used a commonly used uniform WACC (e.g. 7-8% as used by the IEA), the observed differences between GBL and REG would be smaller for Africa and larger for Western Europe.

Related to this point, I think Figure 6 offers a nice point to discuss. If Africa were on a path to reach net-zero before Western Europe, this would have serious political implications (see e.g. debate on historic responsibility). This could be quite brief as I think the discussion and conclusion are currently quite long and could be shortened if the authors wish to.

I would welcome it if the manuscript acknowledged some caveats more clearly. While I understand the reviewers' motivation to show results up to 2100, I still believe that it is worth noting that models over such time horizons typically use uniform WACCs in the notion of a social discount rate because predicting WACCs into the future is notoriously uncertain (cf. movements following the financial crisis or COVID-19). The manuscript should mention this as a note of caution. Moreover, I also understand the authors' justification for the use of corporate finance input data. Depending on the response to the first point (i.e. what sectors are used to estimate input data), I would again suggest to mention the caveat that e.g. when estimating WACCs from samples including upstream companies, the values may not fully represent what is relevant when deploying electricity generation capacity.

Finally, I have a few small remarks concerning language. First, the authors use the terminology of green versus "brown" finance. Recent debates around the BLM have demonstrated that using "brown" to denote dirty and inherently "bad" energy systems may be against good practise of

using inclusive language (see e.g. here: <https://www.bloomberg.com/news/articles/2020-07-24/in-climate-change-brown-is-always-bad-perpetuating-racism?sref=5tCPVKBC>). In this vein, I would encourage the authors to rethink the use of the term. Second, the authors confuse “capital cost” with “cost of capital” at least once (p. 10, 283). This should be checked throughout. Third, the manuscript is now very clear about the use of corporate finance data, but sometimes still mentions project specific financing conditions or risks, which should be avoided (e.g. p. 21, 525).

Response to reviewer 2

Reviewer #2 (Remarks to the Author):

The authors have provided a substantially revised manuscript. I particularly appreciate the new methods section, which gives very helpful detail about the WACC calculation, the model choice and the implementation in TIAM-UCL.

In response to the authors' answer to "WACC input data", I commend the authors for the added clarity regarding WACC data and the type of finance. The methods now clearly lay out how WACC values are derived. However, unless I missed it, there is still a crucial point missing. I would encourage the authors to explain how the green and "brown" WACC values were derived. Table M7 makes the technology classification in TIAM-UCL transparent, which is great. A similar table would be helpful for the derivation of the WACCs, i.e. what sectors were used to estimate betas etc. to obtain the green and "brown" values.

We apologise for missing this aspect in the revised version. The "green" and "brown" WACC values are derived from the Damodaran dataset (2016), which provides beta, equity and country risk premiums, as well as debt-ratio and equity-ratio values at industry level, namely renewables and power sector; while further detail at technology level is not available. We have added this note in the Method section.

In response to the authors' answer to "Charts". I understand the argument against showing all scenarios in one figure. However, I think the observation that GBL and FAST yield (by definition) very similar results for Africa should be stated in the manuscript. This is an interesting point to discuss as it means that models using uniform rates may unintentionally assume ambitious policies. Furthermore, I would still be interested in an explanation (or an intuition) for the bump in Figure 4b, which is in 2040. The authors have provided an intuition for a 2030 effect, but I think this does not explain what we see in the figure.

We thank the reviewer for the suggestion, which is now incorporated in the scenario presentation. Concerning the "bump" in figure 4b, it appears in a panel where changes are presented in term of %; and calculated as difference from the regional scenario (REG). A change in green electricity generation from a small reference level creates this distortion under lower WACC conditions. Note that the level of green electricity generated is increasing in all the scenarios; there are no reduction in green electricity generation between 2040 and 2050 for example in SLOW.

I can understand the authors' argument for weighing values by GDP when aggregating national or regional WACCs. But by definition, this approach leads to a large difference for Africa when comparing GBL to REG and a small one for Western Europe because the former WACC input values "count less" than the latter in calculating the global average. Hence, to some extent, the conclusions the authors reach are driven by this weighing assumption, which is something that needs to be stated clearly. If, for example, the authors used a commonly used uniform WACC (e.g. 7-8% as used by the IEA), the observed differences between GBL and REG would be smaller for Africa and larger for Western Europe.

Related to this point, I think Figure 6 offers a nice point to discuss. If Africa were on a

path to reach net-zero before Western Europe, this would have serious political implications (see e.g. debate on historic responsibility). This could be quite brief as I think the discussion and conclusion are currently quite long and could be shortened if the authors wish to.

We thank the reviewer for the suggestion on potential effects of the GDP weighting, which is now incorporated in the discussion section. Regarding the other point (Africa reaching net-zero before Western Europe), TIAM is a cost optimisation model which goes always to the cheapest option (including financing) under specific constraint (e.g. temperature goal) and our scenarios do not account for net-zero goals set in laws by developed economies. Indeed, in our scenarios while Africa will reach net-zero in 2070 (REG) and in 2060 (FAST), no particular changes are observed for Western Europe that reaches net-zero in 2070 under all different assumptions/scenarios. Cost of financing for power technologies is similar in all scenarios in the EU region but reduced (by several percent points) for Africa in FAST and SLOW scenarios.

I would welcome it if the manuscript acknowledged some caveats more clearly. While I understand the reviewers' motivation to show results up to 2100, I still believe that it is worth noting that models over such time horizons typically use uniform WACCs in the notion of a social discount rate because predicting WACCs into the future is notoriously uncertain (cf. movements following the financial crisis or COVID-19). The manuscript should mention this as a note of caution. Moreover, I also understand the authors' justification for the use of corporate finance input data. Depending on the response to the first point (i.e. what sectors are used to estimate input data), I would again suggest to mention the caveat that e.g. when estimating WACCs from samples including upstream companies, the values may not fully represent what is relevant when deploying electricity generation capacity.

We thank the reviewer for the comment. We have now explicitly mentioned these caveats, namely the uncertainty surrounding WACC values over long-term horizon and WACC values at sectoral level, in "the cost of capital in our scenarios" section.

Finally, I have a few small remarks concerning language. First, the authors use the terminology of green versus "brown" finance. Recent debates around the BLM have demonstrated that using "brown" to denote dirty and inherently "bad" energy systems may be against good practise of using inclusive language (see e.g. here: <https://www.bloomberg.com/news/articles/2020-07-24/in-climate-change-brown-is-always-bad-perpetuating-racism?sref=5tCPVKBC>). In this vein, I would encourage the authors to rethink the use of the term. Second, the authors confuse "capital cost" with "cost of capital" at least once (p. 10, 283). This should be checked throughout. Third, the manuscript is now very clear about the use of corporate finance data, but sometimes still mentions project specific financing conditions or risks, which should be avoided (e.g. p. 21, 525).

We thank the reviewer for these suggestions, which are now incorporated in the main text. In particular, "brown" has been replaced with "high carbon" and "green" with "low-carbon", while the capital cost expression has been corrected with "the cost of capital". We also corrected for any reference to project specific financing conditions.